# Type II taste cells participate in mucosal immune surveillance

**Yumei Qin**[1,2], **Salin Raj Palayyan**[3], **Xin Zheng**[4], **Shiyi Tian**[1], **Robert F. Margolskee**[2], **Sunil K. Sukumaran**[2,3]*

**1** School of Food Science and Bioengineering, Zhejiang Gongshang University, Hangzhou, Peoples Republic of China, **2** Monell Chemical Senses Center, Philadelphia, Pennsylvania, United States of America, **3** Department of Nutrition and Health sciences, University of Nebraska- Lincoln, Lincoln, Nebraska, United States of America, **4** State Key Laboratory of Oral Diseases & National Clinical Research Center for Oral Diseases, Department of Cariology and Endodontics, West China Hospital of Stomatology, Sichuan University, Chengdu, China

* ssukumaran2@unl.edu

**Data Availability Statement:** All sequencing data used in this study can be retrieved from NCBI's short read archive (SRA) with the accession number SRP094673. All other relevant data are

## Abstract

The oral microbiome is second only to its intestinal counterpart in diversity and abundance, but its effects on taste cells remains largely unexplored. Using single-cell RNASeq, we found that mouse taste cells, in particular, sweet and umami receptor cells that express taste 1 receptor member 3 (*Tas1r3*), have a gene expression signature reminiscent of Microfold (M) cells, a central player in immune surveillance in the mucosa-associated lymphoid tissue (MALT) such as those in the Peyer's patch and tonsils. Administration of tumor necrosis factor ligand superfamily member 11 (TNFSF11; also known as RANKL), a growth factor required for differentiation of M cells, dramatically increased M cell proliferation and marker gene expression in the taste papillae and in cultured taste organoids from wild-type (WT) mice. Taste papillae and organoids from knockout mice lacking *Spib* (*Spib^KO^*), a RANKL-regulated transcription factor required for M cell development and regeneration on the other hand, failed to respond to RANKL. Taste papillae from *Spib^KO^* mice also showed reduced expression of NF-κB signaling pathway components and proinflammatory cytokines and attracted fewer immune cells. However, lipopolysaccharide-induced expression of cytokines was strongly up-regulated in *Spib^KO^* mice compared to their WT counterparts. Like M cells, taste cells from WT but not *Spib^KO^* mice readily took up fluorescently labeled microbeads, a proxy for microbial transcytosis. The proportion of taste cell subtypes are unaltered in *Spib^KO^* mice; however, they displayed increased attraction to sweet and umami taste stimuli. We propose that taste cells are involved in immune surveillance and may tune their taste responses to microbial signaling and infection.

## Introduction

In mammals such as the mouse, most taste buds are found in three types of taste papillae on the dorsal surface of the tongue [1,2]. Among them, the fungiform papillae (FFP) located on the anterior tongue each house a single taste bud, and the foliate (FOP) and circumvallate

within the paper and its Supporting Information files.

**Funding:** This study was supported by NIH-NIDCD grants R01 DC014105 and P30 DC011735 to R.F. M., a Young Scientists Fund of the National Natural Science Foundation of China (31800875) to Y.Q. and a Project leader award from the Nebraska Center for the Prevention of Obesity Diseases (NIH-NIGMS grant no. P20GM104320) to S.K.S. Imaging was performed at the Monell Histology and Cellular Localization Core, which is supported, in part, by funding from NIH-NIDCD Core Grant P30DC011735 and National Science Foundation Grant DBI-0216310 (to Gary Beauchamp). The funders had no role in study design, data collection and analysis, decision to publish, or preparation of the manuscript.

**Competing interests:** The authors have declared that no competing interests exist.

**Abbreviations:** APC, antigen-presenting cell; CVP, circumvallate papillae; EGFP, enhanced green fluorescent protein; FACS, fluorescence-activated cell sorting; FAE, follicle-associated epithelium; FFP, fungiform papillae; FOP, foliate papillae; GP2, glycoprotein 2; GST, glutathione-S transferase; IPTG, isopropyl-D-1-thiogalactopyranoside; LB, Luria-Bertani; LPS, lipopolysaccharide; MALT, mucosa-associated lymphoid tissue; M cells, Microfold cells; MSG, monosodium glutamate; NF-κB, nuclear factor kappa B; NT, non-taste; PBS, phosphate-buffered saline; PGLYRP1, peptidoglycan recognition protein 1; PMT, photomultiplier tube; PVR, poliovirus receptor; scRNASeq, single-cell RNASeq; Tas1r3, taste 1 receptor member 3; TLR, Toll-like receptor; TNFSF11, tumor necrosis factor ligand superfamily member 11; WT, wild-type.

(CVP) papillae located laterally and medially on the posterior tongue host a few hundred taste buds each [1,2]. The taste buds in CVP and FOP are arranged around trenches that extend down into the tongue. Each taste bud contains approximately 50 to 100 mature receptor cells classified as type I, type II, type III, and type IV cells based on morphology and marker gene expression [1–3]. These cells are further classified into functional subtypes that respond to basic taste qualities of sweet, bitter, umami (subtypes of type II cells), sour, and salty (mostly subtypes of type III cells) [1–3]. Taste cells have a half-life ranging from 8 to 24 days and are continually regenerated from a population of stem cells located at the base of taste papillae [4–7].

16s RNA sequencing and metagenomics studies have shown that the oral cavity including the tongue dorsum is colonized by a diverse array of microbial species [8–11]. Unlike cells in the non-taste (NT) lingual epithelium, taste cells are continually exposed to the oral microbiota through their microvilli that project to the lingual surface through taste pores. However, their effects on taste signaling and taste cell regeneration have not received the deserved level of attention. The taste papillae are patrolled by a diverse population of immune cells, mostly dendritic cells including the Langerhans cell subtype, T cells, and macrophages [12,13]. Most oral mucosa-resident microbes are harmless or beneficial, and the host develops immune tolerance toward them [14]. However, oral dysbiosis can cause taste loss or distortion, commonly seen in patients with conditions such as influenza, oral thrush (candidiasis), HIV, bacterial infection, and COVID-19 [15–21]. Identifying the pathways underlying the finely tuned crosstalk between taste cells, the oral microbiome and epithelial immune cells will help uncover how microbiota influence taste cells in health and disease.

The most important component of adaptive immunity at the mucosae is the mucosa-associated lymphoid tissue (MALT) [22,23]. MALTs are immune inductive sites that sample the luminal microbes and generate an appropriate mucosal immune response. MALT consists of lymphoid follicles containing B and T cells, which are overlaid by an epithelial layer called follicle-associated epithelium (FAE) [24]. Specialized epithelial cells in the FAE called Microfold cells (M cells) transcytoses luminal microbes and pass them on to antigen-presenting cells (APCs) such as dendritic cells and macrophages housed in their basal pocket [25,26]. Antigen processing and presentation by APCs stimulate the B and T lymphocytes in the underlying lymphoid follicles that mount an appropriate immune response [24,26,27]. These effector cells then migrate to other parts of the mucosae and systemically in blood. Thus, M cells play a central role in mucosal immunity. M cells express several receptors for microbes such as glycoprotein 2 (GP2), peptidoglycan recognition protein 1 (PGLYRP1), and the poliovirus receptor (PVR) that bind to and internalize luminal microbes [28–30]. They have a well-developed microvesicular system, but poorly developed lysosomes enabling rapid transport of their microbial cargo mostly intact across the epithelium [26]. M cell differentiation is induced by tumor necrosis factor ligand superfamily member 11 (TNFSF11; also called RANKL) secreted by connective tissue cells underlying the MALT epithelium [31]. RANKL binds to the receptor tumor necrosis factor receptor superfamily member 11A (TNFRSF11A), which activates the noncanonical nuclear factor kappa B (NF-κB) signaling pathway to turn on the expression of early M cell marker genes [31–34]. The most prominent among them is *Spib*, a transcription factor that orchestrates the later stages of M cell differentiation [35,36]. Using single-cell RNA-Seq (scRNASeq) of GFP-labeled mouse taste cells, we found that taste cells including sweet and umami receptor cells that express taste 1 receptor member 3 (*Tas1r3*) (*Tas1r3*+ cells) express several M cell marker genes [37]. Consistent with this gene expression profile, RANKL administration led to marked up-regulation of M cell marker genes in the taste papillae and cultured taste organoids from wild-type (WT) but not *Spib* knockout (*Spib*$^{KO}$) mice. *Spib*$^{KO}$ mice also showed reduced expression of NF-κB signaling pathway components and

proinflammatory cytokine gene expression in their taste papillae and attracted fewer immune cells to the papillae. However, lipopolysaccharide (LPS)-induced expression of cytokines was highly up-regulated in $Spib^{KO}$ mice compared to their WT littermates. Using a fluorescently labeled microbead uptake assay, we show that taste cells from WT but not $Spib^{KO}$ mice are capable of transcytosis of luminal microparticles. $Spib$ ablation did not affect the proportion of taste cell subtypes in the papillae but caused increased attraction to sweet and umami taste stimuli. Our results indicate that taste cells possess M cell–like properties and might modulate taste signaling in response to microbial colonization and infection.

## Results

### Taste cells express M cell marker genes

To identify genes involved in mucosal immunity expressed in taste cells, we analyzed scRNA-Seq data from $Gnat3$-EGFP-expressing ($Gnat3$-EGFP+, primarily bitter taste receptor expressing in CVP, type II), $Tas1r3$-EGFP+ (sweet and umami receptor expressing, type II), and $Gad1$-EGFP+ (type III) taste cells from CVP of respective EGFP transgenic mice. Our analysis revealed that all cell types, especially the $Tas1r3$+ cells selectively expressed several genes critical for M cell maturation and function (S1 Table) [37]. The expression of a subset of these genes at the mRNA and protein levels in taste cells was confirmed using molecular and histological techniques. Endpoint PCR and quantitative real-time PCR showed robust expression of M cell marker genes $Gp2$, $Marcksl1$, $Ccl9$, $Anxa5$, $Sgne1$, and $Spib$ in the CVP, while they were either expressed at much lower levels or not at all in the NT lingual epithelium (S1A and S1B Fig). RNAscope hybridization with an $Spib$-specific probe set showed that it is expressed in the taste cells in CVP and FFP (S1C and S1D Fig). These results were confirmed at the protein level using indirect immunohistochemistry with an antibody against SPIB, which stained the nuclei of subpopulations of taste cells in CVP, FOP, and PP (S1E and S1G Fig). The specificity of the SPIB antibody and secondary antibody was confirmed by lack of staining in the PP and CVP of $Spib^{KO}$ mice and when the primary antibody was omitted while staining the CVP of WT mice (S1H and S1J Fig).

To identify the taste cell types that express M cell marker genes, we turned to RNAscope Hiplex Fluorescent Assay and double label immunohistochemistry. RNAscope assay was done with probe sets against the M cell marker genes $Spib$, $Gp2$, and $Tnfrsf11a$ and taste cell marker genes $Tas1r3$, $Gnat3$, $Trpm5$ (marker for all type II cells), and $Ddc$ (marker for all type III cells) in the CVP. In agreement with the scRNASeq data, strong colocalization was observed between $Spib$ and $Tas1r3$ with 95% of $Spib$+ cells coexpressing $Tas1r3$ (Fig 1A and S2 Table). About 27% of $Spib$+ cells coexpressed $Gnat3$, 93% coexpressed $Trpm5$, and about 18% coexpressed $Ddc$ (Fig 1B–1D and S2 Table). Notably, the expression level of $Spib$ in $Ddc$+ and $Gnat3$+ cells (indicated by the number of fluorescent spots per cell) is much weaker than in $Tas1r3$+ and $Trpm5$+ cells (Fig 1A–1D). $Gp2$ expression is distributed among all cell types tested, although it tended to be more strongly associated with type II taste cells (Fig 1E–1H and S2 Table). $Tnfrsf11a$ expression is evenly distributed across all cell types (Fig 1I–1L and S2 Table).

Double label immunohistochemistry was done using an SPIB antibody in combination with a second antibody against type II taste cell markers TRPM5, T1R3, and GNAT3, the type III marker CAR4, or the type I marker ENTPD2 (Fig 2) in the CVP and FOP. Our results confirmed that about 91% to 97% of SPIB-expressing cells coexpress TRPM5 and T1R3 (S3 Table). SPIB is weakly coexpressed (8% to 28%) with GNAT3 and negligibly (4% to 6%) with CAR4. SPIB appears to be not coexpressed with ENTPD2, although this was not quantified because type I cells wrap around other taste cells, making accurate determination of coexpression difficult.

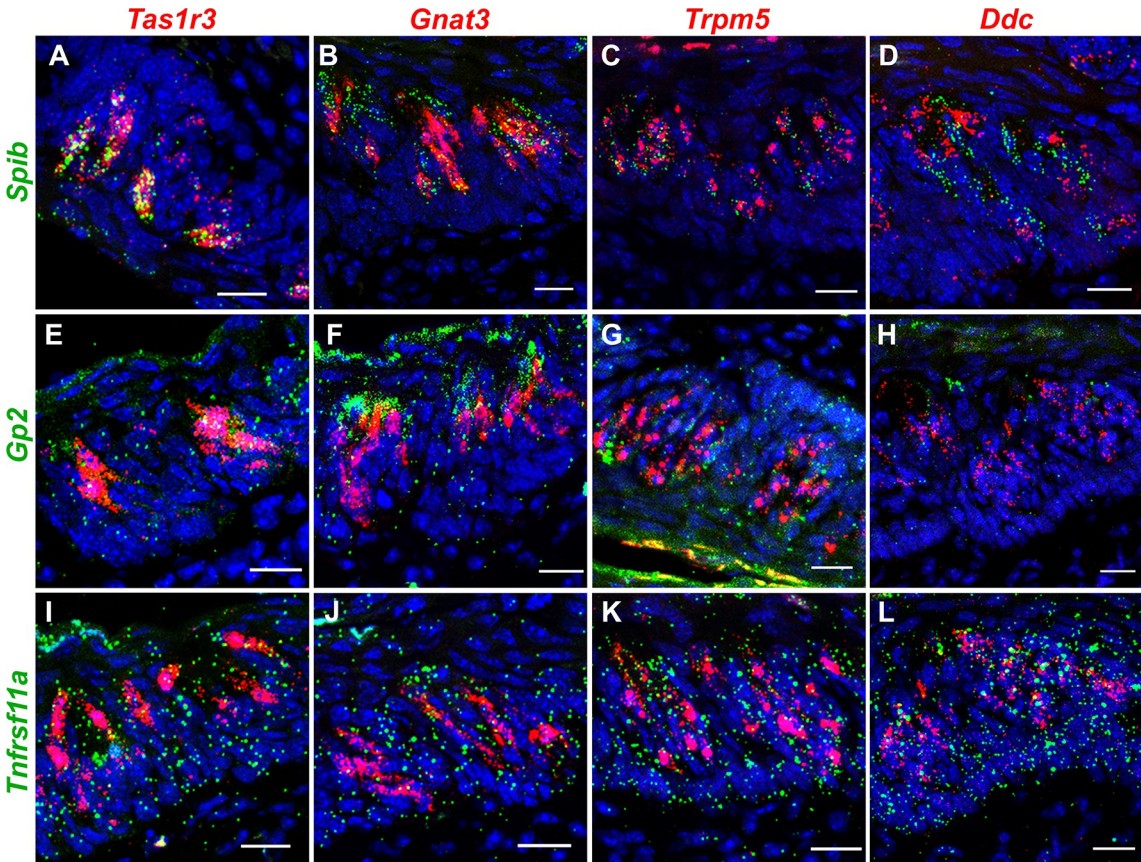

**Fig 1. RNAScope analysis of M cell marker gene expression in taste cells.** RNAscope Hiplex fluorescence assay was used to determine the coexpression of *Spib* (**A-D**), *Gp2* (**E-H**), and *Tnfrsf11a* (**I-L**) with the taste cell markers *Tas1r3*, *Gnat3*, *Trpm5*, and *Ddc* in the CVP. Strong coexpression of *Spib* is observed with *Tas1r3* and *Trpm5* and less strong coexpression was observed with *Gnat3* and *Ddc*. *Gp2* tended to me more correlated with type II taste cells, while *Tnfrsf11a* expression is evenly distributed among all taste cell types. Scale bars = 10 μm.

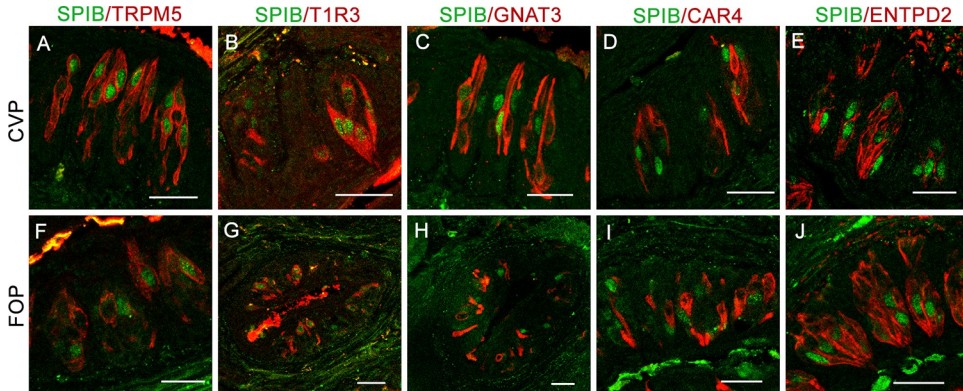

**Fig 2. SPIB is coexpressed with T1R3.** Double-labeled immunofluorescence confocal microscopy of CVP and FOP sections with antibodies against SPIB and type II taste cell markers TRPM5 (**A, F**), T1R3 (**B, G**), and GNAT3 (**C, H**), the type III taste receptor marker CAR4 (**D, I**), and the type I taste cell marker ENTPD2 (**E, J**) show frequent coexpression of SPIB with T1R3 and less frequently with TRPM5 and GNAT3 and negligible coexpression with CAR4 and ENTPD2. Nuclei are counterstained blue with DAPI. Scale bar, 50 μm.

Double label immunohistochemistry also showed that two other M cell markers, GP2 and CCL9, appear to be coexpressed with the type II cell marker TRPM5, but not with the Type III cell marker CAR4 in the CVP, although the coexpression was not quantified (S2 Fig). Of note, the strong GP2 staining at the apex of taste buds, presumably in taste cell microvilli, is similar to the pattern observed in M cells in other MALT for this receptor protein (S2C and S2F Fig) [28]. CVP from *Pou2f3* knockout mice that lack all type II taste cells including *Tas1r3*+ cells do not express CCL9, GP2, and SPIB (S3 Fig) [38]. Collectively, the results above conclusively show that taste cells, in particular, the type II subtype, express several M cell marker genes.

## RANKL up-regulates M cell marker gene expression in taste papillae and cultured taste organoids *Spib* dependently

We next asked if RANKL can trigger *Spib*-dependent increase in expression of M cell marker genes in taste papillae like it does in other MALT [31,35,36]. The basal expression levels of M cell marker proteins GP2, CCL9, MARCKSL1, and SPIB are comparatively low in the WT animals, and the mRNAs encoding *Gp2* and *Ccl9* are lower in CVP of *Spib^{KO}* mice (S4A, S4D, S4I and S4K Fig). Administration of RANKL led to a dramatic increase in the proportion of GP2, CCL9, MARCKSL1, and SPIB expressing cells and the levels of corresponding mRNAs in the CVP of WT but not *Spib^{KO}* mice (S4E and S4L Fig). At the same time, qPCR analysis showed that the expression of *Gp2*, *Ccl9*, and *Marcksl1* mRNAs were not up-regulated by RANKL treatment in the NT epithelium from both strains (S4M Fig).

In the PP and intestinal villi, RANKL treatment induces expression of various M cell marker genes in a stereotypical temporal order [35]. To determine if this is conserved in taste cells, we turned to taste organoids derived from *Spib^{KO}:Lgr5*-EGFP knockin or control (Lgr5-EGFP) strains. Like in PP and intestinal organoids, expression of the early M cell markers *Marcksl1* and *Ccl9* mRNAs and the corresponding proteins started increasing by day 1 and reached their peaks at day 2; that of *Spib* increased gradually and reached the peak by day 3; and that of the late marker *Gp2* increased more slowly and reached its peak at day 4 after RANKL administration in organoids derived from control mice (S5A–S5P Fig). Taste organoids derived from *Spib^{KO}* mice, on the other hand, showed a dramatically impaired induction of *Gp2*, *Ccl9*, and *Marcksl1 mRNAs* upon RANKL administration (S5M–S5P Fig).

## Mucosal immune responses are impaired in *Spib^{KO}* mice

Next, we compared various aspects of mucosal immunity in CVP of WT and *Spib^{KO}* mice. Multiple stimuli, including RANKL, other TNF family ligands, and pathogen-associated molecular patterns such as LPS signal through the NF-κB pathway [39]. We asked if *Spib^{KO}* mice show defects in expression of the components of the NF-κB signaling pathway and its target genes. qPCR analysis showed that the expression of the adapter protein *Myd88*, regulator proteins *Tollip*, *Irak3*, and *Irak4*, and the transcription factors *Nfkb1*, *Nfkb2*, *Rela*, *Rel*, and *Irf6* are down-regulated in *Spib^{KO}* mice (S6A and S6B Fig). In agreement with these findings, expression of several proinflammatory cytokines regulated by NF-κB, namely, *Il1b*, *Il6*, *Mcp1*, and *Tnf* is down-regulated, while that of the anti-inflammatory cytokines *Il10* and *Il12* and *Ifnγ* did not change in *Spib^{KO}* mice (S6C Fig). LPS is known to cause robust proinflammatory cytokine expression in CVP, and we tested if this is recapitulated in *Spib^{KO}* mice [20]. In contrast to baseline expression levels noted above, LPS administration triggered exaggerated cytokine gene expression in CVP of *Spib^{KO}* mice compared to WT littermates (S7 Fig). Next, we asked if the reduced baseline cytokine expression in *Spib^{KO}* mice affected the recruitment of immune cells to the CVP. Immunostaining with antibodies against CD11B and CD45, which label a broad spectrum of immune cells or the T cell marker CD3, showed that *Spib^{KO}* mice

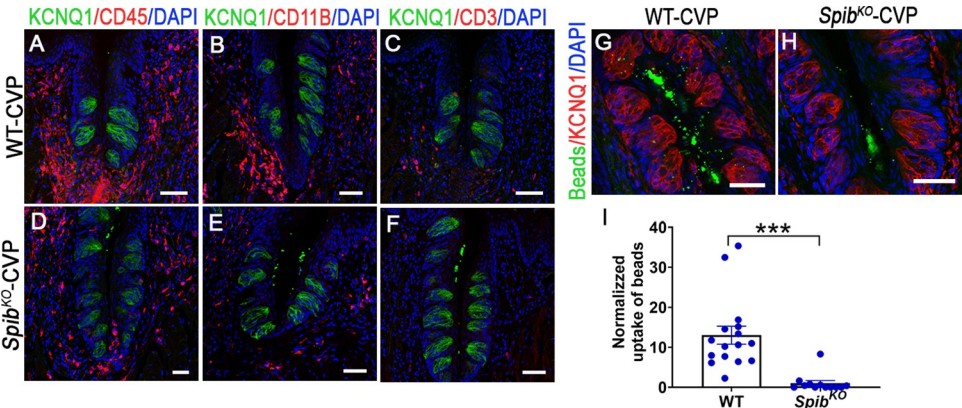

**Fig 3. *Spib* knockout mice have impaired immune responses.** (**A**-**F**) Indirect immunofluorescence confocal microscopy of CVP sections from WT and *Spib^KO* mice stained with antibodies against mouse immune cell markers CD45 (**A**, **D**), CD11B (**B**, **E**), and CD3 (**C**, **F**). Compared to WT mice, *Spib^KO* mice had fewer immune cells in the CVP. (**G**-**H**) Uptake of 200-nm diameter fluorescent beads in taste cells from CVP of WT (**G**) and *Spib^KO* mice (**H**) were observed using confocal microscopy. Taste cells are visualized by staining with a pan-taste cell marker KCNQ1 in panels **A**-**H**. (**I**) Uptake of beads was quantitated by image analysis and normalized so that the average uptake in WT mice was 1.0. (The data underlying the graphs can be found in Data I in S6_Data.) Compared to WT mice, *Spib^KO* mice took up fewer beads. ***$p < .001$. Scale bars, 50 μm.

had fewer immune cells patrolling the CVP (Fig 3A–3F). Finally, we tested if taste cells are capable of microbial transcytosis and if this is impaired in *Spib^KO* mice. The uptake of fluorescently labeled microbeads, a proxy for transcytosis is evident in KCNQ1-expressing taste cells in CVP from WT, but not *Spib^KO* mice (Fig 3G–3I).

## *Spib^KO* mice have enhanced attraction to sweet and umami tastants

Given its role in regeneration and function of M cells, we asked if ablation of *Spib* cause defects in taste cell regeneration and/or taste preference. Immunostaining with antibodies against the taste marker proteins T1R3, TRPM5, GNAT3, and CAR4 showed that the proportion of type II and III cells are unaltered in the CVP of *Spib^KO* mice compared to WT (S8A–S8I Fig). These results were confirmed by qPCR analysis of the corresponding mRNAs (S8J Fig). However, in brief access taste tests, *Spib^KO* mice displayed increased attraction to the prototypical sweet and umami taste stimuli sucrose and monopotassium glutamate compared to WT littermates (Fig 4A and 4C). The attraction to sucralose appeared to be higher as well in *Spib^KO* mice, although not statistically significant (Fig 4B). On the other hand, the responses to denatonium benzoate (bitter), sodium chloride (salty), and citric acid (sour) that are mediated by taste cell types other than *Tas1r3+* cells were unchanged between the two strains (Fig 4D–4F).

## Discussion

Mucosae are important routes of microbial colonization, and animals have evolved a strong mucosal immune system comprised of both innate and adaptive components to counter infection [40–42]. The MALT is a type of secondary lymphoid tissue critical for mucosal adaptive immunity. The MALT in the gut (PP) and tonsils are the best studied, but MALT occurs in other mucosae such as the salivary glands, nasopharynx, conjunctiva, and tear ducts as well [24,43,44]. The mucosa in the tongue dorsum is heavily colonized by the oral microbiome, even more so than the buccal mucosa [9,45]. Most of the tongue surface is made up of keratinized stratified squamous epithelium that acts as an effective barrier to microbial invasion. However, the microvilli of taste cells project to the tongue surface through the taste pores in

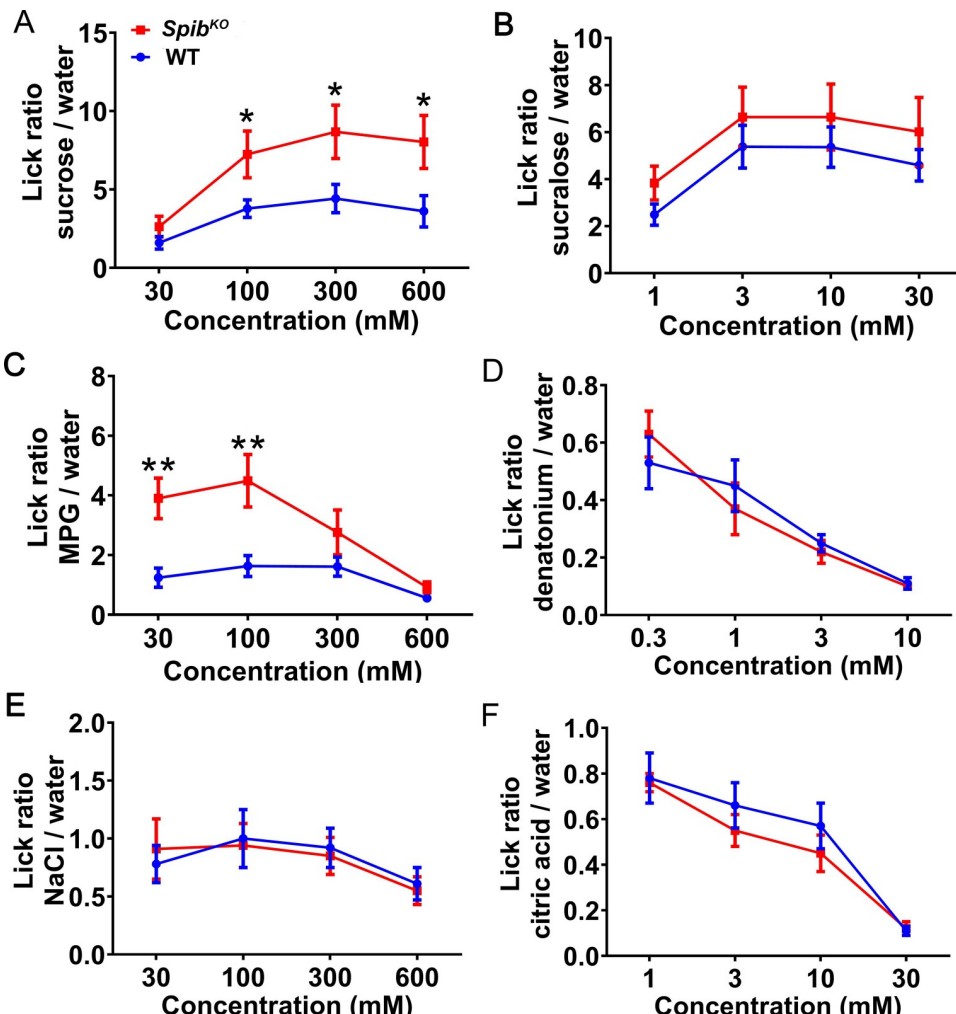

**Fig 4. *Spib^KO* mice show increased behavioral attraction to sweet and umami tastants.** Brief access tests were used to measure behavioral responses to sweet (sucrose and sucralose, **A** and **B**), umami (monopotassium glutamate [MPG], **C**), bitter (denatonium, **D**), salty (NaCl, **E**), and sour (citric acid, **F**) taste stimuli. (The data underlying the graphs can be found in Data A-F in S8_Data.) Compared to littermate control WT mice, *Spib^KO* mice show increased lick responses to sweet and umami (sucrose and MPG), while the responses to other taste stimuli are unchanged. Lick ratios were calculated by dividing the number of licks to a taste solution by the number of licks to water in each test session. Data are means ± SEM analyzed with two-way ANOVA with post hoc *t* test. $N = 12$ (WT mice) and 14 (*Spib^KO* mice). $^*p < .05$, $^{**}p < .01$.

the taste buds and presumably represent an easier route for microbial invasion. The trenches in the CVP and FOP surrounded by taste buds are ideal sites for long-term microbial colonization, as they are largely shielded from salivary flushing. However, the effects of the oral microbiota on taste cells have not been studied in sufficient detail so far. The expression of Toll-like receptors (TLRs), interferon receptors, and their downstream signaling pathway components in taste cells has been documented [46–48]. Administration of LPS or double-stranded RNA that binds to TLRs and mimics bacterial and viral infection, respectively, activates TLR and interferon signaling pathways in taste cells and diminishes taste cell regeneration and taste transduction, likely by promoting secretion of proinflammatory cytokines [20,47,49,50]. Similarly, knocking out *Tlr4*, the primary receptor for LPS in mice, leads to diminished taste response to sugars, lipids, and umami [48]. Interestingly, sweet and bitter taste receptors bind

microbial metabolites and mediate immune responses in several extra oral tissues, but whether they do so in taste papillae itself is not known [51,52].

There are several functional and developmental similarities between taste and intestinal epithelia. For instance, intestinal epithelial cells and taste cells in the CVP and FOP are of endodermal origin and arise from *Lgr5*-expressing stem cells [4,53,54]. Taste receptors and other members of the taste transduction machinery are also expressed in the nutrient-sensing enteroendocrine cells and the microbe- and parasite-sensing tuft cells in the intestine [55–60]. Finally, both taste and intestinal epithelial cells are heavily exposed to the respective microbiota, and it is plausible that some taste cell types may share functional features of M cells. Indeed, we identified an M cell–like gene expression signature in taste cells using scRNASeq (Figs 1, 2, S1, and S2 and S1 Table). Strikingly, RANKL administration led to M cell proliferation in CVP and up-regulation of M cell marker genes in cultured taste organoids in the same temporal order observed in the PP and PP-derived organoids (S4 and S5 Figs). Finally, as in the PP, M cell marker gene expression in CVP and taste organoids requires *Spib*. *Spib*$^{KO}$ mice have low basal levels of M cell gene expression and fail to respond to RANKL (S4 and S5 Figs). In agreement with these observations, taste cells from WT but not *Spib*$^{KO}$ mice are able to transcytose luminal microbeads (Fig 3G–3I). Thus, all evidence indicates that taste cells have M cell–like properties. However, it is not clear if all taste cell types have properties of M cells. In the basal level, type II cells, in particular the *Tas1r3*+ population, strongly coexpress *Spib* and *Gp2* at both mRNA and protein levels (Figs 1, 2, and S2). The expression level of *Spib* (as determined by the number of fluorescent spots per cell) in *Ddc*+ cells is much lower than in type II cells (Fig 2A–2H). However, it is unlikely that all the cells expressing M cell marker genes upon RANKL stimulation are type II cells or if only type II cells are capable of transcytosis. Some type III taste cells also express *Tnfrsf11a* (RANKL receptor), and it possible that they form a sizeable proportion of cells expressing M cell marker genes upon RANKL stimulation (Fig 1L and S1 and S2 Tables). Similarly, it is possible that type III taste cells are also capable of transcytosis, although a subtype-level transcytosis assay was not done due to technical difficulties. Nevertheless, the basal level expression of SPIB, GP2, and CCL9 is abrogated in *Pou2f3*$^{KO}$ animals, indicating a prominent role for type II taste cells in immune surveillance (S3 Fig). It is unlikely that type I or IV cells are capable of transcytosis since their microvilli do not project out of the taste pore [3]. In summary, our data indicate that type II taste cells, in particular, *Tas1r3*+ cells, likely form the bulk of M cell–like taste cells, although it is possible that a smaller subset of other taste cells, likely type III cells can also acquire M cell properties.

Despite the similarities outlined above, there are several key differences between taste cells and classical M cells. Taste cells do not have the basal pocket found in M cells that houses APCs. In this respect, they are analogous to villous M cells that spontaneously transdifferentiate from enterocytes in apices of intestinal villi [61,62]. Similarly, villous M cells and the taste papillae are not associated with underlying germinal centers [63]. Villous M cell–stimulated T and B cell maturation may occur in the intestinal lamina propria, but it is not known if this occurs in the lamina propria of taste papillae [27]. However, taste papillae from WT mice contain a diverse population of immune cells, and *Spib*$^{KO}$ mice have far fewer of them, strongly supporting a role for taste cells in immune surveillance (Fig 3A–3F). Of note, although not present in rodents, humans and most other mammals have lingual tonsils located near the CVP and FOP. They contain MALT, where taste cell-stimulated T and B cells may mature, although this might happen in the cervical or other lymph nodes as well. In fact, the mesenteric lymph nodes appear to be the primary sites of T and B cell maturation in intestinal mucosa [64].

What other roles might the M cell–like properties of taste cells serve? Taste cells secrete cytokines such as IL-10 and TNF alpha, which affect taste signaling [49,50,65]. M cells are

known to regulate secretion of cytokines in the PP[66]. $Spib^{KO}$ mice express lower levels of proinflammatory cytokines and have fewer immune cells in the taste papillae while the expression of anti-inflammatory cytokines is unaltered (Figs 3A–3F and S6C). The expression of components of the NF-κB signaling pathway that regulate acute phase cytokine gene expression is lower in CVP of $Spib^{KO}$ mice, which might underlie this phenotype (S6A and S6B Fig). Of note, increased T cell recruitment to taste papillae was observed in IL-10 knockout mice, which also had smaller taste buds and fewer taste cells per bud [65]. The lower levels of immune cell recruitment in CVP of $Spib^{KO}$ mice likely reflects down-regulation of cytokines, especially chemokines (Figs 3A–3F and S6C). However, upon LPS stimulation, $Spib^{KO}$ mice show exaggerated inflammatory cytokine gene expression (S7 Fig). These data and the inability of taste cells in $Spib^{KO}$ mice to transcytose luminal particles indicate that mucosal immune responses are impaired in the CVP in the absence of antigen surveillance. $Spib$ is required for the development and regeneration of M cells and plasmacytoid dendritic cells [35,36,67]. Strikingly, the proportion of taste cell subtypes in $Spib^{KO}$ mice is unaltered, suggesting that its role in taste cells is likely restricted to regulating expression of genes involved in mucosal immunity (S8 Fig). A direct role in regulating taste signaling itself remains an open question. The higher behavioral attraction to sweet and umami stimuli in $Spib^{KO}$ mice could arise from reduced expression of proinflammatory cytokines or changes in expression of downstream components of the taste signaling pathway (Fig 4). Our findings indicate that taste cell–mediated immune surveillance is a key aspect of oral mucosal immunity and that the dysregulation of this pathway may lead to microbial infection and taste loss.

## Materials and methods

### Animals

All animal experiments were performed in accordance with the National Institutes of Health guidelines for the care and use of animals in research and reviewed and approved by the Institutional Animal Care and Use Committee at Monell Chemical Senses Center (protocols: 1163 and 1151 to RFM). Animals were housed in a specific pathogen-free vivarium with a 12-h light/dark cycle and open access to food and water. $Spib^{KO}$ mice on a C57BL/6J background were obtained from Dr. Lee Ann Garrett-Sinha (State University of New York, Buffalo, USA) [68]. Lgr5-EGFP-IRES-CreERT2 mice (stock #008875) were purchased from the Jackson Laboratory. $Tas1r3$-GFP, $Gad1$-GFP, and $Gnat3$-GFP transgenic mice were as previously described [69–71]. Lgr5-EGFP-IRES-CreERT2 mice were crossed with $Spib^{KO}$ mice to obtain Lgr5-EGFP-IRES-CreERT2:$Spib^{KO}$ and Lgr5-EGFP-IRES-CreERT2:$Spib^{+/+}$ animals for generating taste organoids. Genotypes were confirmed using primer sets recommended by Jackson Laboratory and the Sinha lab.

### RANKL preparation and injection

The expression and purification of recombinant mouse RANKL were performed as previously described with some modifications [31]. Truncated mouse RANKL transcript (encoding amino acids 137–316 of mouse RANKL) was PCR amplified (forward primer: CACCCCCG GGCAGCGCTTCTCAGGAGCT, reversed primer: GAGACTCGAGTCAGTCTA TGTCC TGAAC) and then cloned into pGEX-5X-2 vector (GE Healthcare) between SmaI and XhoI sites. The insert was verified by sequencing and then transformed into BL21 $Escherichia coli$ strain (Stratagene) for expression. Bacteria harboring expression plasmids were selected and grown in Luria-Bertani (LB) media supplemented with 100 μg/ml ampicilin. The cultures were induced with 20 μM isopropyl-D-1-thiogalactopyranoside (IPTG) for 16 h at 25˚C, and the glutathione-S transferase (GST)-tagged RANKL (GST-RANKL) were purified from bacterial

lysate by affinity chromatography on a GSTrap FF column (Cat. No. 71-5016-96 AM, GE Healthcare, Chicago, IL) followed by dialysis against multiple changes of phosphate-buffered saline (PBS) (pH 7.4). Recombinant GST used as a control was prepared by the same method using empty pGEX-5X-2. GST or GST-RANKL, with at least 95% purity as demonstrated by SDS–PAGE, was administered to *Spib^KO* and their littermate control animals by intraperitoneal injections (250 μg/day/mice for 3 days). Mice were killed at day 4 and lingual epithelium or lingual tissue were collected.

**LPS treatment of mice.** Half of WT and *Spib^KO* mice received a single intraperitoneal injection of 5 mg/kg LPS (dissolved in PBS), and the other half received an injection of vehicle (PBS buffer) as control. Mice were killed 6 h after LPS treatment for extraction of taste tissue.

**Isolation of lingual epithelium.** Lingual epithelium was enzymatically peeled as described [37]. Mice were killed by $CO_2$ asphyxiation, and the tongues were excised. An enzyme mixture (0.5 ml) consisting of dispase II (2 mg/ml; Roche, Mannheim, Germany; cat. no. 04942078001) and collagenase A (1 mg/ml; Roche cat. no. 10103578001) in $Ca^{2+}$-free Tyrode's solution (145 mM NaCl, 5 mM KCl, 10 mM HEPES, 5 mM $NaHCO_3$, 10 mM pyruvate, 10 mM glucose) was injected under the lingual epithelium and incubated for 15 min at 37˚C. Lingual epithelia were peeled gently from the underlying muscle tissue and used for fluorescence-activated cell sorting (FACS) or RNA isolation.

**Fluorescence-activated cell sorting.** GFP-fluorescent *Lgr5+* taste cells from Lgr5-EGFP-IRES-CreERT2; *Spib^KO* and Lgr5-EGFP-IRES-CreERT2;*Spib^{+/+}* mice were isolated by FACS as described [4,72]. Briefly, the region of the lingual epithelium containing the CVP from 3 mice was excised and pooled, minced into small pieces, incubated with trypsin (0.25% in PBS) for 10 to 25 min at 37˚C, and mechanically dissociated into single cells using heat-pulled Pasteur pipettes. Cell suspensions were filtered using 70-μm cell strainers (BD Biosciences, Bedford, MA; cat. no. 352350) and then with 35-μm cell strainers (BD Biosciences cat. no. 352235). Cells were sorted into culture medium for organoid culture, based on the enhanced green fluorescent protein (EGFP) signal (excitation, 488 nm; emission, 530 nm).

**3D taste organoid culture.** Taste organoids were cultured as described [4,72]. Briefly, FACS-sorted GFP fluorescent cells were mixed with 4% chilled Matrigel (v/v; BD Biosciences, San Jose, CA; cat. no. 354234) and maintained in DMEM/F12 (Thermo Fisher, Waltham, MA; cat. no. 11320–033) supplemented with HEPES (10 mM, Thermo Fisher cat. No. 15630080), GlutaMAX (2 mM, Thermo Fisher; cat. No. 35050061), Wnt3a-conditioned medium (50%, v/v), R-spondin-conditioned medium (20%, v/v), Noggin-conditioned medium (10%, v/v), N2 (1%, v/v; Thermo Fisher; cat. No. 17502–048), B27 (2%, v/v; Thermo Fisher; cat. No. 12587–010), Y27632 (10 μM; Sigma-Aldrich; cat. No. Y0503), and epidermal growth factor (50 ng/mL; Thermo Fisher). Wnt3a- and R-spondin-conditioned medium were generated from Wnt3a and R-spondin stable cell lines as described [73]. Noggin conditioned medium was made in house. The culture medium was changed first at days 5 to 7 and once every 2 to 3 days thereafter. After 14 days culturing, recombinant mouse RANKL was added into the fresh culture medium at concentration of 50 ng/ml for 5 consecutive days. At various time points (day 0 to day 5), taste organoids were harvested.

**PCR and qPCR.** Total RNA was isolated taste papillae or NT epithelium dissected from freshly peeled lingual epithelium or taste organoids using the PureLink mini kit (Thermo Fisher; cat. no. 12183018A) with on-column DNA digestion (using PureLink DNase, Thermo Fisher; cat. no. 12185010) and converted into cDNA using Super Script VILO kit (Thermo Fisher; cat. no. 11755050). PCR and qPCR were done as previously described. All qPCR results were normalized using the ΔΔCt method with *Gapdh* as reference. Primers used are shown in S4 Table.

**Tissue preparation.** Mice were killed by cervical dislocation and taste papillae-containing portions of the tongue were quickly removed and briefly rinsed in ice-cold PBS. For

RNAScope assay, tissues were freshly frozen in Tissue-Tek O.C.T. mounting media (Sakura) using a 100% ethanol dry ice bath and then sectioned within 1 h after dissection. For immunohistochemistry, tissues were fixed for 1 h at 4°C in 4% paraformaldehyde/1× PBS and cryoprotected in 20% sucrose/1× PBS overnight at 4°C before embedding in O.C.T. Sections (8 to 10 μm thickness) were prepared using a CM3050S cryostat (Leica Microsystems) and applied on precoated microscope slides (Superfrost plus; Fisher Scientific). Sections used for RNAScope were immediately stored at −80°C and those for immunohistochemistry were dried at 40°C for 20 min before storage at −80°C.

**RNAScope Hiplex assay.** RNAscope assay was done using the Hiplex12 fluorescent assay kit for mouse (Catalog # 324443, Advanced Cell Diagnostics, Hayward, CA, USA) per manufacturer's instructions. Positive and negative control probes were run in parallel to test probes to ensure proper hybridization and imaging conditions were attained in our experiments. Confocal images were captured using a Nikon A1R-Ti2 confocal microscope. Z-series stack with 6 images per stack was captured at a step size of 1 μm. Images were scanned using a 512 × 512 pixel format, scan lines were averaged twice, and frames were scanned 3 times. Acquisition parameters [i.e., gain, offset, photomultiplier tube (PMT) setting] were held constant for experiments. Colocalization counts were made using QuPath software from Advanced Cell Diagnostics. Cell boundaries were detected automatically based on DAPI staining. The fidelity of each cell boundary was confirmed by manual inspection. The number of fluorescent spots in each channel (that corresponds to individual mRNA molecules) per cell were extracted and used for colocalization counting. Data from more than 2 nonconsecutive sections from 2 mice were pooled.

**Immunostaining.** Immunostaining of sections containing CVP, FOP, or FFP papillae was done as described [37,74]. Briefly, sections were rinsed in PBS and blocked with SuperBlock blocking buffer (Thermo Fisher; cat. no. 37515) supplemented with 0.3% (v/v) Triton X-100 and 2% donkey serum for 30 min at room temperature. Sections were incubated overnight with primary antibodies at 4°C. After 3 × 5 min washing with 1× PBS, species-specific secondary antibodies were used to visualize specific taste cell markers. 6-Diamidino-2-phenylindole (DAPI; 1:1,000) in deionized water was used to visualize the nuclei following secondary antibody staining. The primary and secondary antibodies used in this study are listed in S5 Table.

Immunostaining of taste organoids was done as previously described [4,72]. Briefly, cultured organoids were collected through centrifuging and fixed for 15 min in fresh 4% paraformaldehyde in 1× PBS supplemented with $MgCl_2$ (5 mM), EGTA (10 mM), and sucrose (4%, wt/v). After 3 × 5 min washing with 1× PBS, organoids were blocked for 45 min with SuperBlock blocking buffer (Thermo Fisher cat. no. 37515) supplemented with 0.3% (v/v) Triton X-100 and 2% (v/v) donkey serum and then incubated at 4°C overnight with the desired primary antibodies. They were washed 3× for 5 min with 1× PBS and incubated for 1 h with species-specific secondary antibodies (1:500) at room temperature. 6-Diamidino-2-phenylindole (DAPI, 1:1,000) in deionized water was used to visualize the nuclei following secondary antibody. All images were captured with the TCS SP2 spectral confocal microscope (Leica Microsystems). Scanware Software (Leica Microsystems) was used to acquire z-series stack captured at a step size of 0.5 μm. Images were scanned using a 512 × 512 pixel format, scan lines were averaged twice, and frames were scanned 3 times. Acquisition parameters [i.e., gain, offset, PMT setting] were held constant for experiments with antibodies and for controls without antibodies. Digital images were cropped and arranged using Photoshop CS (Adobe Systems). Colocalization for each taste cell marker was done using images from at least 2 sections showing entire CV papillae or foliate papillae of each mouse was counted ($N$ = 3). Only those taste cells for which the entire cell body and nucleus could be visualized were counted.

**Assessment of the uptake of fluorescent beads by taste cells.** Mice anesthetized by an intraperitoneal injection (10 ml/kg) of a mixture of ketamine (4.28 mg/ml), xylazine (0.86 mg/

ml), and acepromazine (0.14 mg/ml) were administered fluorescent polystyrene latex nano-particles (Fluoresbrite YG; 200 nm diameter) (Polysciences, Warrington, PA; cat. No. 09834–10). After 2 h, tongues were isolated and fixed using 4% paraformaldehyde in 1× PBS. Then, frozen sections of 10 μm thickness were obtained as described above and stained with an anti-body specific for T1R3. Confocal images were captured as described above. Quantitative analysis of fluorescent beads uptake from lingual epithelium containing CVP was done as previously described [31]. Analysis of the degree of bead uptake from oral cavity containing CVP by threshold analysis using ImageJ. Images of the fluorescent beads were saved as 8-bit grayscale images and then converted to binary images. The percentage of the pixels with a signal intensity that exceeded the threshold cutoff point of 75 out of 255 was calculated for the area occupied by CVP. Images of DAPI-staining nuclei in the same field acquired in a separate channel were threshold at a cutoff point of 70. The extend of bead uptake was expressed as the ratio of pixels with fluorescent beads to pixels with DAPI after normalization to a mean value of 1.0 for CVP from WT mice. Taste buds (the region of taste papillae excluding the trench region and the lamina propria) from 2 or 3 sections with entire CVP of each mouse was counted ($N = 5$).

**Brief access taste tests.**    Brief access tests were conducted using the Davis MS-160 mouse gustometer (Dilog Instruments, Tallahassee, FL) as described [72,74,75]. The following taste compounds were tested: sucrose (30, 100, 300, 1,000 mM), sucralose (1, 3, 10, 30 mM), mono-sodium glutamate (MSG; 30, 100, 300, 600 mM), denatonium (0.3, 1, 3, 10 mM), citric acid (1, 3, 10, 30 mM), and NaCl (30, 100, 300, 600 mM). Mice were water- and food-restricted (1 g food and 1.5 mL water) for 23.5 h before test sessions for appetitive taste compounds (sucrose, sucralose, and MPG). For the aversive taste compounds (citric acid, denatonium, and NaCl), mice were water- and food-deprived for 22.5 h before testing. In each test session, 4 different concentrations of each taste compound and water control were presented in a random order for 5 s after first lick, and the shutter reopened after a 7.5-s interval. The total test session time was 20 min. An additional 1-s "washout period" with water was interposed between each trial in sessions testing aversive tastants. $Spib^{KO}$ and their littermates $Spib^{+/+}$ mice were tested at the same time in parallel. Each mouse was tested with all the compounds. After each session, mice were allowed to recover for 48 h with free access to food and water. Body weight of the mice was monitored daily, and the mice at or over 85% their initial were used. The ratio of taste stimulus to water licks was calculated by dividing the number of licks for taste compounds by the number of licks for water presented in the parallel test session. Lick ratios >1 indicate pref-erence behavior to the taste compound, and lick ratios <1 indicate avoidance behavior to the taste compound.

**Statistical analyses.**    Prism (GraphPad Software) was used for statistical analyses, includ-ing calculation of mean values, standard errors, and unpaired $t$ tests of cell counts and qPCR data. Data from taste behavioral tests were compiled using Microsoft Excel. For statistical anal-yses of behavioral responses, two-way ANOVA and post hoc pairwise multiple comparisons with Tukey's correction were used to evaluate the difference between genotype and concentra-tion using the tidyverse and emmeans packages in R. The level of statistical significance was set at $p < .05$. $p$-Values $< .05$ were considered significant for paired $t$ tests of qPCR results. All data used for statistical analyses are provided in the Supporting information table. The code used for this analysis is provided in the Supporting information file S1 Code.

## Supporting information

**S1 Data. Underlying data and results of statistical analyses reported in the figures in the main text and Supporting information files.** The figures and the sheets in the excel file

corresponding to them are as follows: S1 Fig = S1_Data, S4 Fig = S2_Data, S5 Fig = S3_Data, S6 Fig = S4_Data, S7 Fig = S5_Data, Fig 3 = S6_Data, S8 Fig = S7_Data, Fig 4 = S8_Data. Data for each panel in figures are labeled with the corresponding panel label.
(XLSX)

**S1 Code. R Code for analysis of gustometer data.** The code for analysis of gustometer data in Fig 4 for sucrose is shown below. The codes for other tastants were similar except for corresponding changes in file names and concentrations.
(DOCX)

**S1 Fig. Expression of *Spib* and other M cell marker genes in taste cells.** (**A**) End point PCR (35 cycles) of *Gapdh* (housekeeping control gene) *Gnat3* (taste tissue control gene), *Gp2*, *Ccl9*, *Marcksl1*, *Anxa5*, *Sgne1*, and *Spib* from cDNA prepared CVP, non-taste lingual epithelium (NT), and Peyer's Patch (PP). *Gp2*, *Ccl9*, *Sgne*1, and *Spib* are expressed in CVP and PP, but not in NT. *Marcksl1* and *Anxa5* are expressed in all tissues, although at lower levels in the NT. Lines on the right indicate position of molecular weight markers. (**B**) qPCR analysis of above genes expression in CVP and NT confirms that all M cell marker genes are highly expressed in CVP, while they are expressed at lower levels or not at all in NT. The expression of each marker gene is plotted as the logarithm of the ratio between its cycle threshold values to those of *Gapdh*. Individual datapoints from each cDNA sample in shown in blue dots (the underlying data can be found in Data B in S1_Data). (**C**-**D**) RNAscope hybridization using an *Spib*-specific probe set produced strong signals in subsets of taste cells in CVP and FFP. (**E**-**G**) Indirect immunofluorescence confocal microscopy of cryosection from taste papillae and PP stained with an SPIB antibody shows strong nuclear staining in subpopulations of taste cells in CVP, FOP, and PP. (**H**-**I**) Absence of SPIB signal in CVP and PP of *Spib*$^{KO}$ mice proves the specificity of SPIB antibody. (**J**) Omission of the primary antibody (W/O) demonstrates low nonspecific background from secondary antibody in CVP. Nuclei are counter stained with DAPI in panels C-J. Scale bars, 50 μm. $^{**}p < .01$, $^{***}p < .001$.
(TIF)

**S2 Fig. Coexpression of M cell and type II taste cell marker proteins.** Double-labeled immunofluorescence confocal microscopy of CVP sections with antibodies against M cell markers GP2 and CCL9, with the type II taste cell marker TRPM5 (**A**-**C** and **G**-**I**) or the type III taste receptor marker CAR4 (**D**-**F** and **J**-**L**) in the CVP. Merged images show GP2 and CCL9 are coexpressed with TRPM5 (**B**, **H**), but not CAR4 (**E**, **K**). Nuclei are counterstained blue with DAPI. Scale bar, 50 μm.
(TIF)

**S3 Fig. M cell marker genes are not expressed in *Pou2f3* knockout mice lacking type II taste cells.** Immunofluorescence images with antibodies against SPIB (**A**), GP2 (**B**), and CCL9 (**C**) show no staining in CVP from *Pou2f3* knockout mice that lack all type II taste cells. Nuclei are counterstained blue with DAPI. Scale bar, 50 μm.
(TIF)

**S4 Fig. RANKL stimulates M cell marker gene expression through Spib in CVP.** (**A**-**H**) Indirect immunofluorescence confocal microscopy of CVP sections from WT mice treated or untreated using RANKL stained with antibodies against M cell markers GP2, SPIB, CCL9, and MARCKSL1. The result showed that administration of RANKL led to a dramatic increase in the proportion of taste cells expressing these proteins. (**I**-**K**) qPCR analysis showed that all the above marker genes and *Spib* were significantly up-regulated after RANKL treatment in WT but not *Spib*$^{KO}$ mice. ND, not done. (**M**) RANKL treatment did not affect the expression level

of M cell marker genes in NT lingual epithelium. Data are means ± SEM. (The data underlying the graphs can be found in Data I-M in S2_Data.) $^{**}p < .01$, $^{***}p < .001$. Nuclei are counter-stained blue with DAPI. Scale bar, 50 μm.
(TIF)

**S5 Fig. RANKL treatment induces expression of M cell marker genes in a stereotypical temporal order in cultured taste organoids.** (**A-L**) Indirect immunofluorescence confocal microscopy of taste organoids from control and *Spib*$^{KO}$ mice showing kinetics of expression of the M cell markers MARCKSL1 (**A-D**), CCL9 (**E-H**), and GP2 (**I-L**) 0–3 days after RANKL treatment. (**M-P**) qPCR analysis of the expression of M cell markers in cultured taste organoids after RANKL treatment. The kinetics of expression of GP2, CCL9, and MARCKSL1 after RANKL treatment were distinct at both protein (**A-L**) and mRNA (**M-P**) levels. *Spib*$^{KO}$ mice failed to up-regulate all M cell marker genes (**M-P**) upon RANKL administration. (The data underlying the graphs can be found in Data M-P in S3_Data.) Data are mean means ± SEM. Scale bars, 50 μm.
(TIF)

**S6 Fig. *Spib*$^{KO}$ mice have impaired immune responses.** qPCR analysis of proinflammatory cytokines and the components of NF-κB signaling pathway in WT and *Spib*$^{KO}$ mice. (**A-B**) The expression of the transcription factors *Nfkb1*, *Nfkb2*, *Rela*, Relb, *Rel*, *Irf6*, and regulator proteins *Tollip*, *Myd88*, *Irak3*, and *Irak4* belonging to the NF-κB signaling pathway and *Irf6* are significantly down-regulated in *Spib*$^{KO}$ mice. (**C**) Compared to WT mice, *Spib*$^{KO}$ mice showed lower expression of proinflammatory cytokines *Il1b*, *Il6*, *Tnf*, and *Mcp*, but the expression of anti-inflammatory cytokines *Il10* and *Il12* did not change. (The data underlying the graphs can be found in Data A-C in S4_Data.) Data are mean means ± SEM. $^{*}p < .05$.
(TIF)

**S7 Fig. qPCR analysis of the LPS-induced expression of inflammatory cytokines.** Compared to WT mice, LPS administration triggered exaggerated cytokine expression in CVP of *Spib*$^{KO}$ mice. (The data underlying the graphs can be found in Data A-F in S5_Data.) Data are means ± SEM. $^{*}p < 0.05$, $^{**}p < .01$, $^{***}p < .001$.
(TIF)

**S8 Fig. The proportion of taste cell types are unaltered in *Spib*$^{KO}$ mice.** (**A-H**) Indirect immunofluorescence confocal microscopy of CV sections from WT and *Spib*$^{KO}$ mice immunostained for type II cells using markers T1R3 (**A**, **E**), TRPM5 (**B**, **F**), GNAT3 (**C**, **G**), and type III taste cells with antibodies against CAR4 (**D**, **H**). Nuclei are counterstained with DAPI (blue). (**I**) Compared to WT mice, the proportion of taste receptor cells in *Spib*$^{KO}$ mice were unaltered. (The data underlying the graphs can be found in Data I in S7_Data.) (**J**) qPCR analysis of expression of the corresponding genes confirms this observation (Data J in S7_Data). Data are mean means ± SEM with individual data points from replicates shown as black/blue dots. Scale bar, 50 μm.
(TIF)

**S1 Table. Expression of M cell marker genes in taste cells.** Normalized scRNASeq data from T1r3-GFP (Tas1-10), Gnat3-GFP (Gnat 1–10), and Gad1-GFP (Gad1-12) cells are shown. The statistical comparison shown was done between T1r3-GFP and Gad1-GFP cells using DESeq2.
(XLSX)

**S2 Table. Coexpression of *Spib*, *Gp2*, and *Tnfrsf11a* mRNAs with mRNAs encoding taste marker genes.** RNAscope Hiplex assay was done in CVP with probe sets against *Spib*, *Gp2*, *Tnfrsf11a*, *Tas1r3*, *Gnat3*, *Trpm5*, and *Car4*, and singly and doubly labeled cells were counted.

Numerators are the numbers of taste cells expressing both gene 1 and gene 2. Denominators are the numbers of taste cells expressing gene 1. Taste cells expressing both gene 1 and gene 2 as a percentage of those expressing gene 1 are shown in parentheses. ND, not determined. (DOCX)

**S3 Table. Coexpression of SPIB with taste marker genes.** Taste cells from CVP and FOP were doubly stained for SPIB and a second antibody against T1R3, GNAT3, TRPM5, or CAR4, and singly and doubly labeled cells were counted. Numerators are the numbers of taste cells expressing both gene 1 and gene 2. Denominators are the numbers of taste cells expressing gene 1. Taste cells expressing both gene 1 and gene 2 as a percentage of those expressing gene 1 are shown in parentheses. ND, not determined. (DOCX)

**S4 Table. Nucleotide sequences of primers used for RT-PCR and qPCR experiments.** (DOCX)

**S5 Table. Primary and secondary antibodies used for immunofluorescence experiments.** *RRID, Research Resource Identifier. (DOCX)

**S1 Raw Image. Raw image showing all the wells in the agarose gel used to analyze RT-PCR data shown in S1A Fig.** (PDF)

# Acknowledgments

The authors wish to thank Dr. Lee Garrett-Sinha at the State University of New York at Buffalo for providing the *Spib* knockout mouse strain, Dr. Lynn Corcoran at the Walter and Eliza Hall Institute for Medical Research, Victoria, Australia, for the SPIB antibody, Dr. Hong Wang for creative suggestions, and Mr. Kevin Redding for technical assistance.

# Author Contributions

**Conceptualization:** Yumei Qin, Robert F. Margolskee, Sunil K. Sukumaran.

**Data curation:** Yumei Qin, Salin Raj Palayyan, Xin Zheng, Shiyi Tian, Sunil K. Sukumaran.

**Formal analysis:** Yumei Qin, Salin Raj Palayyan, Xin Zheng, Shiyi Tian, Robert F. Margolskee, Sunil K. Sukumaran.

**Funding acquisition:** Robert F. Margolskee, Sunil K. Sukumaran.

**Investigation:** Yumei Qin, Salin Raj Palayyan, Xin Zheng, Shiyi Tian, Sunil K. Sukumaran.

**Methodology:** Yumei Qin, Salin Raj Palayyan, Sunil K. Sukumaran.

**Project administration:** Yumei Qin, Robert F. Margolskee, Sunil K. Sukumaran.

**Resources:** Robert F. Margolskee, Sunil K. Sukumaran.

**Supervision:** Robert F. Margolskee, Sunil K. Sukumaran.

**Validation:** Yumei Qin, Salin Raj Palayyan.

**Visualization:** Yumei Qin, Salin Raj Palayyan.

**Writing – original draft:** Yumei Qin, Sunil K. Sukumaran.

**Writing – review & editing:** Yumei Qin, Salin Raj Palayyan, Robert F. Margolskee, Sunil K. Sukumaran.

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
