## [Editor Report · Decision Letter 0]

22 Apr 2022

Dear Dr Sukumaran, 

Thank you for submitting your manuscript entitled "Sweet taste receptor cells may participate in mucosal immune surveillance" for consideration as a Research Article by PLOS Biology.

Your manuscript has now been evaluated by the PLOS Biology editorial staff as well as by an academic editor with relevant expertise and I am writing to let you know that we would like to send your submission out for external peer review. However, we would like to consider the manuscript as a Short Report, thus please select this type of article from the dropdown menu when you submit the metadata (see below).

Before we can send your manuscript to reviewers, we need you to complete your submission by providing the metadata that is required for full assessment. To this end, please login to Editorial Manager where you will find the paper in the 'Submissions Needing Revisions' folder on your homepage. Please click 'Revise Submission' from the Action Links and complete all additional questions in the submission questionnaire.

Once your full submission is complete, your paper will undergo a series of checks in preparation for peer review. Once your manuscript has passed the checks it will be sent out for review. To provide the metadata for your submission, please Login to Editorial Manager (https://www.editorialmanager.com/pbiology) within two working days, i.e. by Apr 25 2022 11:59PM.

If your manuscript has been previously reviewed at another journal, PLOS Biology is willing to work with those reviews in order to avoid re-starting the process. Submission of the previous reviews is entirely optional and our ability to use them effectively will depend on the willingness of the previous journal to confirm the content of the reports and share the reviewer identities. Please note that we reserve the right to invite additional reviewers if we consider that additional/independent reviewers are needed, although we aim to avoid this as far as possible. In our experience, working with previous reviews does save time. 

If you would like to send previous reviewer reports to us, please email me at ialvarez-garcia@plos.org to let me know, including the name of the previous journal and the manuscript ID the study was given, as well as attaching a point-by-point response to reviewers that details how you have or plan to address the reviewers' concerns. 

Kind regards,

Ines

--

Ines Alvarez-Garcia, PhD

Senior Editor

PLOS Biology

---

## [Decision Letter · Decision Letter 1]

31 May 2022

Dear Dr Sukumaran,

Thank you for your patience while your manuscript entitled "Sweet taste receptor cells may participate in mucosal immune surveillance" was peer-reviewed at PLOS Biology as a Short Report. It has now been evaluated by the PLOS Biology editors, an Academic Editor with relevant expertise, and by three independent reviewers. 

As you will see, the reviewers find your conclusions novel and potentially interesting, however they also raise several issues that would need to be thoroughly addressed before we can consider the manuscript for publication. The reviewers think that you should perform several experiments and immunostainings to strengthen the conclusions, and that some of the results are overinterpreted and need to be toned down or better justified. One of the main issues is the confirmation that the sweet taste receptor cells specifically mediate oral immune responses.

In light of the reviews and discussions with the Academic Editor, we would like to invite you to revise the work to thoroughly address the reviewers' reports.

Given the extent of revision needed, we cannot make a decision about publication until we have seen the revised manuscript and your response to the reviewers' comments. Your revised manuscript is likely to be sent for further evaluation by all or a subset of the reviewers.

**IMPORTANT - SUBMITTING YOUR REVISION**

3. Resubmission Checklist

a) *PLOS Data Policy*

b) *Published Peer Review*

Sincerely,

Ines

--

Ines Alvarez-Garcia, PhD

Senior Editor

PLOS Biology

Reviewers' comments

Rev. 1:

These authors demonstrate that taste receptor cells express key genes and proteins involved in M cell surveillance of the gut microbiome and might play a similar role in the oral cavity. This is an exciting study that opens the door to novel taste cell-microbial interactions. The work is also timely given the altered oral microbiome in COVID-19 patients and a possible link to taste loss. The authors used compelling mix of in vivo and organoid models. The manuscript is generally well-written and most of my suggestions are relatively minor. Most significant are the lengthy presentation of results in the introduction and questions about interpreting results.

* In the title and throughout the manuscript the focus is on sweet taste receptor cells. While that might be shorthand for sweet- and umami-sensing type II cells I don't find "STRC" accurate given expression and behavioral data.

* The introduction would be improved by summarizing the main findings more concisely.

* Methods

1) Are mice housed in a SPF barrier facility which relates to microbial exposure?

2) What is the route and dose of LPS administration?

3) Data were analyzed by two-way ANOVAs followed by t tests as multiple comparisons. Were P values adjusted to control Type I error introduced by multiple t tests?

4) Provide details for confocal imaging including objective, zoom and optical thickness of z scans.

* Figure 1L. What are the heavily stained SPIB+ cells at the base of foliate papillae? If they are glands this might be discussed later in comparison to the gut.

* Figure 2. The two lower magnification panels make it difficult to see co-localization vs. background.

* Page 8 line 181. The statement that "Collectively, the results above conclusively show that STRCs express several M cell marker genes" in not precise. Some of the genes were co-localized with TRPM5/type II markers rather than T1R3.

* Figure S2 and accompanying text/legend. The results demonstrate an increase in M cell marker genes upon treatment with RANKL but not proliferation, which was not directly tested.

* Legends for Fig. S3 and 5 have the same title. It was also not clear why some of the results were presented as supplementary figures rather than included in the main text unless there are journal limits.

* Figure 5. The images showing bead uptake are not very convincing. Were beads at the apical surface of the taste counted? It appears that only 2-3 beads are present within the taste bud.

* To my knowledge, M cells fail to develop from progenitors in SPIB KO mice. In contrast there were no differences in the different types of taste cells in KO mice. The difference in the gut and taste cell phenotypes in SPIB KO mice is striking and could be added to the discussion section comparing and contrasting taste cells and M cells.

Rev. 2:

This manuscript reports that the subset of taste receptor cells that respond to sweet tastes express markers for M cells of the immune system, including SPIB. A comparison between the taste cells and immune cells is further supported by experiments showing that administration of RANKL leads to an increase in the number of cells that express these markers, while expression of these markers is lower in a Spibko mouse strain. Most remarkably, the authors find that Spibko mice show a heightened preference for sucrose and MSG over water, as compared to WT mice. Overall, there are some interesting and potentially very important findings in this manuscript.

Major:

1. Much of the paper rests on immunocytochemistry to define the population of cells that express SPIB. However, because SPIB is nuclear, and the taste cell markers are on the plasma membrane (or cytosol), it is sometimes difficult to tell in which cells they are co-expressed (and how this was assessed by the authors). This is made more difficult by quality of some of the images. One solution would be to use the TRPM5-GFP mouse strain generated by this PI, which should show better colocalization. It would also be helpful if the authors added a Venn diagram to show cell counts (cells in which the proteins are co-expressed, or expressed alone).

2. Orgenoid IHC is not convincing given low resolution images and poor morphology of the cells, and adds little to the manuscript. It could be put in a supplementary figure.

3. Images showing bead uptake are not convincing - the beads appear to be stuck at what appears to be the apical surface of the tastebud (labeling the taste pore would be helpful). The negative control does not have a similar number of T1R3 labelled cells. Did the investigators ever observe beads inside the cells? This should be shown if possible.

4. The increased preference for sucrose and MPG is impressive. But some critical questions need to be answered. First, it is surprising that a similar change in preference is not seen for sucralose. How can this be explained? And without a mechanism, this change in behavior could be due to global effects of the knockout (on M cells throughout the body, leading, for example, to changes in metabolism). To show that there is a change in the peripheral response of taste cells, and connect the results to the main findings of the papers, the authors should, at a minimum, show results from gustatory nerve recording.

5. Is there something about this strain or the way that the animals were tested that makes them so insensitive to ionic tastes? First, the mice continue to lick high concentrations of NaCl (300 mM and 600 mM) that are typically very aversive (at 600 mM, the still lick at >50% of the rate that they lick to H2O). Also, concentrations of citric acid that are typically strongly aversive (30 mM - 300 mM) appear to not be aversive or only mildly aversive, and the concentrations in the figure are different from those in the methods.

Minor:

Fig 1 A - please include MW markers

Fig 1B - the gene "Spi-B" is presumably Spib?

Fig 1C-J - label figure with name of probe Fig K-N - In the absence of taste cell markers, itt is hard to tell where the taste buds are in the IHC, making the expression of SPIB not very convincing. Same for KO. I suggest leaving this out, and simply showing the IHC in Fig 2, along with the KO controls (counterstained for taste cell markers).

Fig 6 A - SpiB knockout - should be Spib? (gene name)

Methods: Was tissue fixed in PFA for 1h or overnight (line 452)?

Rev. 3:

In this manuscript the authors suggest that a subset of sweet sensitive taste receptor cells in mouse taste buds functions in immune surveillance of pathogens in the oral cavity. The mechanistic hypothesis is very exciting and very timely given the impact of SARS-CoV2 on smell and taste function. They interrogate their published scRNAseq dataset of 3 different taste bud cell types for genes expressed by microfold (M) cells that function in other immune tissues and show that several of these are expressed in taste cells. They first use immunostaining to show several microfold cell marker proteins, including Spib, Gp2, and Ccl9, are co-expressed in taste bud cells. They then use organoid technology to activate the immune response with the RANKL, the ligand for Tnfrsf11a, which they find enriched in taste bud cells in the RNAseq dataset, and show substantial upregulation of several M cell gene products in response to RANKL. They further show that Spib is required for this response in organoids and in mice. In vivo, M cells have epithelial transcytosis function and can take up fluorescent beads, and the authors show data that Tas1r3+ cells in mouse taste buds also take up these beads and uptake is Spib-dependent. Finally, they use a behavioral assay to show that compared to controls, Spib-cKO mice have altered taste responses to sweet and umami tastants, but not to bitter, salt or sour, and suggest that in addition to functioning in immune surveillance, sweet taste cells might "might modulate taste preference in response to microbial colonization and infection "

While these are intriguing findings, their data do not appear to fully support a model where sweet taste receptor cells uniquely function to mediate oral immune responses. In fact there data suggest that there are complex expression patterns of M cell genes in taste epithelium and complex responses to immune challenge in taste epithelium that are not consistent with their interpretation.

The authors rely on their published scRNAseq dataset of mouse CVP taste cells, made up of only 32 cells and roughly 10 cells per cell type (GAD1+ type III, Gnat3+ bitter type II and Tas1r3+ sweet and umami type II taste cells). In table S1, the authors present M cell genes enriched in their dataset. Combined with the highly variable expression of each gene within each group, comparisons of differentially regulated genes across groups should be more cautiously interpreted by the authors here. Comparing Tas1r3+ sweet and umami type II cells vs Gad1+ type III cells for example show differences between these 2 populations, however, the authors are making the strong statement that these immune related genes are upregulated specifically in sweet taste receptor cells. Of interest, some genes they suggest are functionally involved in immune response of taste buds are actually most highly expressed in Gad1+ cells, i.e., Gp2, Tnfrsf11a. They include scRNAseq expression data also for the Gnat3+ bitter type II cells. A quick comparison of Gnat3+ and Tas1r3+ cells shows that many of these genes are not upregulated specifically in sweet cells but instead are expressed in both type II populations, e.g. Spib, or more highly, although quite variably, expressed in bitter cells e.g., Scg5, Gp2. PCR and immunofluorescence analysis (Figs 1-3) also suggests that these M cell genes are expressed in a complex pattern among bitter and sweet/umami cells, as well as in the non-taste epithelium of the tongue (Marcksl1, Anxa5). The authors appear to gloss over this complexity:"Using single cell RNASeq of GFP-labeled mouse taste cells, we found that sweet taste receptor cells (STRCs) and a few other taste cell types express several M cell marker genes" It is also important to note that Tas1r3 is also expressed by umami-sensitive taste buds cells, and thus limiting the interpretation of the function of Tas1r3+ cells to sweet taste cells is an oversimplification as well.

Additional experiments that would improve the manuscript would be to show explicit co-expression of M cell genes with Tas1r2 mRNA in taste tissue.

Specific critiques

Immunofluorescence for GP2 or CCL9 in Fig 3 appear to show many cells in taste buds express these proteins, not just TRPM5+ taste cells, which is expressed by bitter, sweet and umami type II cells. In panel 3D-F, GP2 appears to also be expressed in a CAR4+ cell, and the extent to which GP2 is co-expressed in 3A-C is difficult to determine as immunofluorescence is brightest at the taste pore. Given the apparent high background of GP2 and CCL9 the authors should show control tissue with primary antisera omitted, especially since GP2 is a mouse antibody.

Further in Fig S2, GP2 appears not to be expressed in control taste tissue, only in that from mice treated with RANKL, and with RANKL treatment many more cells appear immunopositive for GP2, CCL9 and MARCKSL1, and SPIB than in controls and that can be accounted for by expression specific to only Tas1r3+ sweet/umami cells. The authors should assess these expression patterns with TAS1R3 double labeling.

In Figs 2, 3 and 5 the authors do not indicate if these are projections of confocal z-stacks or optical sections; the latter are required to determine if a single taste cell is double labeled, or if a single cell has taken up a fluorescent bead.In figure 5, how many taste buds and cells were analyzed for bead transcytosis?

The authors should show individual datapoints for plots in Fig 1B, 4M-P, 5I, and 6A-F, S2I-L, S3A-C, S4A-F, S5B-F, .

In table S2, how many mice were used in this analysis?

---

## [Decision Letter · Decision Letter 2]

2 Nov 2022

Dear Dr Sukumaran,

Thank you for your patience while we considered your revised manuscript entitled "Type II taste cells may participate in mucosal immune surveillance" for publication as a Short Report at PLOS Biology. This revised version of your manuscript has been evaluated by the PLOS Biology editors, the Academic Editor and one of the original reviewers. The other reviewer was not currently available, but the Academic Editor has checked your responses.

Based on the reviews (attached below), we are likely to accept this manuscript for publication, provided you satisfactorily address the remaining points raised by Reviewer 3. Please also make sure to address the following data and other policy-related requests stated below.

In addition, we would like you to consider a suggestion to improve the title:

"Type II taste cells participate in mucosal immune surveillance"

We expect to receive your revised manuscript within two weeks. 

*Published Peer Review History*

*Press*

Sincerely,

Ines

--

Ines Alvarez-Garcia, PhD

Senior Editor

PLOS Biology

Fig. 1B; Fig. 5I-M; Fig. 6I; Fig. 7I, J; Fig. 8A-F; Fig. S2M-P; Fig. S3A-C and Fig. S4A-F

**Please also ensure that figure legends in your manuscript include information on WHERE THE UNDERLYING DATA CAN BE FOUND, and ensure your supplemental data file/s has a legend.

We require the original, uncropped and minimally adjusted images supporting all blot and gel results reported in an article's figures or Supporting Information files. We will require these files before a manuscript can be accepted so please prepare and upload them now. Please carefully read our guidelines for how to prepare and upload this data: https://journals.plos.org/plosbiology/s/figures#loc-blot-and-gel-reporting-requirements

Reviewers' comments:

Rev. 3:

The authors have addressed my concerns.

2 minor corrections noted:

Line 237 change "proliferation of M cells" to "increased expression of M cell marker genes"

Line 488 methods. How long after LPS treatment were mice harvested?

---

## [Editor Report · Decision Letter 3]

29 Nov 2022

Dear Sunil,

We are almost ready to accept for publication your manuscript entitled "Type II taste cells participate in mucosal immune surveillance" at PLOS Biology as a Short Report. However, we have realised your manuscript currently doesn't comply with our Short Reports format as it currently has eight main figures instead of four. Please accept my sincere apologies for this oversight during the first revision.

An easy fix at this stage would be to renumber the figures and make four of them supplementary as follows:

Fig. 1: no change

Fig. 2: no change

Fig. 3: no change

Fig. 4: no change

Fig. 5: rename as Fig. S2

Fig. 6: rename as Fig. S6

Fig. 7: rename as Fig. S7

Fig. 8: rename as Fig. S8

Fig. S1: no change

Fig. S2: rename as Fig. S3

Fig. S3: rename as Fig. S4

Fig. S4: rename as Fig. S5

Please upload new copies of the figures with the new numbering and rename also the figures in the text.

We expect to receive your revised manuscript within one week. 

Sincerely,

Ines

--

Ines Alvarez-Garcia, PhD

Senior Editor,

ialvarez-garcia@plos.org,

PLOS Biology

---

## [Editor Report · Decision Letter 4]

10 Dec 2022

Dear Dr Sukumaran,

Thank you for the submission of your revised Short Report entitled "Type II taste cells participate in mucosal immune surveillance" for publication in PLOS Biology. On behalf of my colleagues and the Academic Editor, Piali Sengupta, I am delighted to say that we can in principle accept your manuscript for publication, provided you address any remaining formatting and reporting issues. These will be detailed in an email you should receive within 2-3 business days from our colleagues in the journal operations team; no action is required from you until then. Please note that we will not be able to formally accept your manuscript and schedule it for publication until you have completed any requested changes.

PRESS

Sincerely, 

Ines

--

Ines Alvarez-Garcia, PhD

Senior Editor

PLOS Biology
